# Molecular Characterization of UDP-*N*-Acetylglucosamine Pyrophosphorylase and Its Role in the Growth and Development of the White-Backed Planthopper *Sogatella furcifera* (Hemiptera: Delphacidae)

**DOI:** 10.3390/genes13081340

**Published:** 2022-07-27

**Authors:** Zhao Wang, Gui-Yun Long, Cao Zhou, Dao-Chao Jin, Hong Yang, Wen-Jia Yang

**Affiliations:** 1College of Life and Health Science, Kaili University, Kaili 556011, China; hdwangzhao@126.com; 2Provincial Key Laboratory for Agricultural Pest Management of Mountainous Regions, Institute of Entomology, Guizhou University, Guiyang 550025, China; lgy0256@126.com (G.-Y.L.); zhouc@cqnu.edu.cn (C.Z.); axyridis@163.com (H.Y.); 3College of Life Sciences, Chongqing Normal University, Chongqing 401331, China; 4Guizhou Provincial Key Laboratory for Rare Animal and Economic Insects of the Mountainous Region, College of Biology and Environmental Engineering, Guiyang University, Guiyang 550005, China; yangwenjia10@126.com

**Keywords:** UDP-*N*-acetylglucosamine pyrophosphorylase, *Sogatella furcifera*, gene expression profiling, RNA interference

## Abstract

UDP-*N*-acetylglucosamine pyrophosphorylase (UAP) is a key enzyme in the chitin biosynthesis pathway of insects. Here, we described the gene *SfUAP* in the white-backed planthopper *Sogatella furcifera* (Horváth) with an open reading frame of 1470 bp. Quantitative real-time polymerase chain reaction (qPCR) suggested that *SfUAP* exhibits a different developmental expression pattern and a higher expression after molting. The highest expression of *SfUAP* was observed in the integument tissues of adults, whereas head tissues showed negligible expression. RNAi-based gene silencing decreased the mRNA transcript levels in *S. furcifera* nymphs injected with double-stranded RNA of *SfUAP*. Finally, *SfUAP* silencing led to 84% mortality and malformed phenotypes in nymphs. Thus, our results can help better understand the role of *SfUAP* in *S. furcifera*.

## 1. Introduction

The white-backed planthopper *S. furcifera* is one of the most destructive insect pests of rice plants in some Asia–Pacific countries. Recently, the outbreak frequency of *S. furcifera* has been increasing in China [1,2]. This pest severely reduces rice yields by ovipositing, sucking the rice phloem sap, and serving as a virus vector to transmit southern rice black-streaked dwarf virus [3,4,5]. The application of chemical insecticides has been an effective measure for controlling *S. furcifera*. However, the frequent and irrational use of these insecticides has led to resistance against various insecticides in this pest [6,7]. Therefore, it is urgent to discover more scientific and efficient pest management approaches to control *S. furcifera*.

Chitin, a linear polymer of *N*-acetyl-*β*-d-glucosamine linked by *β*-1,4-glycosidic bonds, is the second most prevalent biological polysaccharide in nature and is found in various organisms, including microorganisms (yeast, mycelial fungi, and algae) [8,9], arthropods (insects, mites, and crustaceans) [8,9,10], other invertebrates (nematodes, mollusks, and sponges) [11,12], and some vertebrates (fish and amphibians) [13,14]. In insects, it plays an essential role in the epidermis, peritrophic matrix (PM), and other internal structures; moreover, chitin combines with sclerosis protein to constitute the cuticular exoskeleton [10,15,16]. A hard exoskeleton can help insects sustain their body shape and defend against external mechanical disruption [17]; however, it restrains insects’ growth and development. For this reason, during insect growth and metamorphosis, chitin must be periodically degraded by chitinase and synthesized via the chitin biosynthesis pathway. Once chitin degradation or formation is disrupted, molting and metamorphosis are blocked, ultimately resulting in insect death [18,19,20,21,22,23]. Further, the chitin biosynthesis pathway comprises several biochemical reactions that are catalyzed via enzymes. In general, the pathway begins with trehalose; involves at least eight enzymes, such as trehalase, hexokinase, glucose-6-phosphate isomerase, glutamine fructose-6-phosphate aminotransferase, glucosamine-6-phosphate-*N*-acetyltransferase, phosphoacetylglucosamine mutase, UDP-*N*-acetylglucosamine pyrophosphorylase (UAP) and chitin synthase; and ends with the chitin polymer [10,24]. To date, several studies have reported on the chitin formation pathway in insects; however, most focused on the first trehalase and the last chitin synthase [25]; knowledge regarding the structural features and roles of other enzymes involved in this pathway remains limited.

UAP (EC 2.7.7.23), a vital regulatory enzyme in the insect chitin biosynthesis pathway, specifically catalyzes the reaction of *N*-acetylglucosamine-1-phosphate with uridine triphosphate to yield UDP-*N*-acetylglucosamine (UDP-GlcNAc), which is an important substrate for the formation of chitin [24,26]. So far, UAPs have been isolated and characterized from several insect species, including *Locusta migratoria* [25], *Bactrocera dorsalis* [27], *Tribolium castaneum* [28], *Spodoptera exigua* [29], *Aedes aegypti* [30], *Cnaphalocrocis medinalis* [31,32], *Leptinotarsa decemlineata* [33], *Bombyx mori* [34], and *Henosepilachna vigintioctopunctata* [35]. Interestingly, most insects have been known to only possess a single *UAP* gene, except *T. castaneum*, *L. migratoria*, and *L. decemlineata*, all of which have two *UAP* genes (*UAP1* and *UAP2*). RNA interference (RNAi) was used to investigate their physiological functions, which demonstrated that silencing *TcUAP1* or *LmUAP1* reduced chitin contents in the integument and PM, and thus arrested insect growth [25,28]. Further, in *L. decemlineata*, silencing *LdUAP1* only decreased chitin contents in the integument, whereas the knockdown of *LdUAP2* resulted in a reduction of chitin contents in the PM [33]. In a previous study on *Drosophila melanogaster*, *DmUAP* mutants (also called *cystic*, *mummy*, or *cabrio*) caused several defects in tracheal tubule and eye development, central nervous system fasciculation, and cuticle formation [36,37,38]. These results strongly suggest that each *UAP* gene plays a different role in insect development and survival. However, the number and function of *UAP* genes in *S. furcifera* have not yet been characterized. Furthermore, the mechanism by which UAP enzyme inhibition affects the regulation of chitin biosynthesis in *S. furcifera* is yet to be investigated.

In this study, we identified and characterized a full-length cDNA of the *UAP* gene in *S. furcifera* and assessed its expression patterns across different tissues and developmental stages. Moreover, we investigated the roles of the *SfUAP* gene by RNAi. These results are helpful in understanding the functions of chitin biosynthesis pathway components in *S. furcifera* and provide a target for developing new biological pesticides.

## 2. Materials and Methods

### 2.1. Rearing S. furcifera

The *S. furcifera* used in this study were initially obtained from rice fields in Guiyang, Guizhou Province, China, in 2013. They had not been exposed to any pesticides for 6 consecutive years and were routinely reared on Taichung Native-1 (TN1) rice seedlings in a climatic chamber at 25 ± 2 °C, with 70 ± 10% relative humidity and a 16:8 h light/day photoperiod. TN1 was grown on soil in a growth incubator at 30 °C ± 2 °C, with a 16:8 h light/day photoperiod. These insects were transferred to fresh rice seedlings every 15–20 days to ensure adequate nutrition.

### 2.2. Total RNA Isolation and cDNA Preparation

Total RNA was extracted from the whole body of *S. furcifera* using TRIzol reagent (Invitrogen, Carlsbad, CA, USA), according to the manufacturer’s instructions. RNA integrity was determined using 1% agarose gel electrophoresis, and their concentration and purity were assessed on a Nanodrop 2000 spectrophotometer (Thermo Fisher Scientific, Wilmington, DE, USA). Before the analysis, all samples were stored at −80 °C. cDNA was synthesized using the AMV First-Strand cDNA Synthesis Kit (Sangon Biotech, Shanghai, China) with an oligodT primer, according to the user manual provided by the manufacturer, and stored at −20 °C until analysis.

### 2.3. Cloning SfUAP Using Reverse Transcription Polymerase Chain Reaction (RT-PCR) and RACE

Based on the partial sequences of *SfUAP* obtained from the transcriptome sequencing data of *S. furcifera* [39], primers were designed via the Primer Premier 6.0 software using cDNA as the template. PCR amplifications were conducted using the primer pairs listed in Table 1 and L.A. Taq^®^ polymerase (TaKaRa, Dalian, China) in 25μL of reaction mixtures containing 2 μL of dNTPs (2.5 mM), 2.5 μL of 10× LA PCR Buffer (Mg^2+^ plus), 1 μL of each primer (10 mM), and l μL of cDNA template. Thermal cycling conditions were as follows: one cycle of predenaturation at 94 °C for 3 min; followed by 30 cycles of denaturation at 94 °C for 30 s, annealing at 52 °C for 30 s, and extension at 72 °C for 2 min; and a final extension at 72 °C for 10 min. Target bands of the amplified products were purified using the EasyPure^®^ Quick Gel Extraction Kit before using 1% agarose gel electrophoresis (Transgen Biotech, Beijing, China). Purified DNA was cloned into a pMD18-T vector (TaKaRa, Dalian, China) and sequenced using the Sangon Biotech sequencer (Shanghai, China). Next, the BLAST analysis of the sequence was conducted using NCBI for further validation, which identified a 770 bp fragment including the 3′ untranslated region.

We used the SMARTer^®^ RACE 5′/3′ Kit (Clontech, Mountain View, CA, USA) to amplify the full-length cDNA of *SfUAP*. Further, 5′ RACE was used to amplify the 5′ end using two nested gene-specific primers (GSPs), namely UAP-51 and UAP-52. The reaction conditions for the primary RACE–PCR with the universal primer mix and GSP were as follows: 35 cycles of denaturation at 94 °C for 30 s, annealing at 52 °C for 30 s, and extension at 72 °C for 60 s. For the nested PCR reaction, we diluted the primary PCR products 100 times, and then used them as templates with the universal primer short and GSP. Notably, the nested PCR program followed the same conditions as the primary PCR program. Following this, the PCR products were purified using the EasyPure^®^ Quick Gel Extraction Kit, subcloned into the pMD18-T vector, and sequenced via Sangon Biotech sequencer.

### 2.4. cDNA and Amino Acid Sequence Analysis

The full-length sequence of the *SfUAP* gene was constructed by assembling the sequencing fragments using SeqMan software. DNAMAN 7.0 (Lynnon Biosoft, California, CA, USA) was used to edit the nucleotide sequence. A similarity search and homology comparison were performed using the NCBI BLAST program (https://blast.ncbi.nlm.nih.gov/Blast.cgi, accessed on 20 July 2019). The NCBI tool “open reading frame (ORF) finder” was used to search the ORF (https://www.ncbi.nlm.nih.gov/orffinder/, accessed on 20 July 2019). The molecular weight and isoelectric point (pI) of amino acids were predicted based on the amino acid sequences deduced using the ProtParam tool at ExPASy (https://www.expasy.org/, accessed on 28 July 2019). The *N*-glycosylation sites were analyzed using the NetNGlyc 1.0 Server (http://www.cbs.dtu.dk/services/NetNGlyc/, accessed on 28 July 2019). The functional domains of the SfUAP protein were predicted online using PROSITE (https://prosite.expasy.org/, accessed on 28 July 2019). The phylogenetic tree was constructed via MEGA 6.06 using the neighbor-joining method, and bootstrap analyses of 1000 replicates were performed to test the topology. The SWISS-MODEL program (https://www.swissmodel.expasy.org/interactive, accessed on 28 July 2019) was used to constitute the homology models of SfUAP, which were then visualized using the PyMOL Molecular Graphics System 1.1.

### 2.5. SfUAP Expression in Different Developmental Stages and Tissues Using Quantitative Real-Time PCR (qPCR)

*S. furcifera* was sampled from various stages ranging from egg to adult to determine the expression of *SfUAP* at different developmental stages. To evaluate tissue-specific expression, tissue samples were collected from the integument, fat body, gut, and head of the first-day fifth-instar nymphs and from the ovary of 3-day-old adults. Three biological replicates were used for each sample. First, total RNA was isolated from each sample using the H.P. Total RNA Kit (with genomic DNA removal columns; Omega bio-tek, Norcross, GA, USA) and then examined using 1% agarose gel electrophoresis to ensure their integrity. The final concentration was assessed using a Nanodrop 2000 spectrophotometer. The first-strand cDNA was synthesized from 2 μg of RNA using the AMV RT reagent Kit (Sangon Biotech) with an oligodT primer. Table 2 lists the GSPs used for qPCR. It was performed in a CFX-96 real-time qPCR system (Bio-Rad, Hercules, CA, USA) with 20μL of reaction volumes containing 10 μL of FastStart Essential DNA Green Master (Roche Diagnostics, Shanghai, China), 1 μL of cDNA, 1 μL (10 mM) of each primer, and 7 μL of RNase-free water. The amplification conditions were as follows: an initial denaturation at 95 °C for 10 min, followed by 40 cycles at 95 °C for 30 s and at 55 °C for 30 s. After the reaction, a melting-curve analysis was performed from 65 °C to 95 °C to confirm the specificity of the PCR results. Our previous evaluations normalized data to the stable reference gene 18S ribosome RNA (GenBank accession no. HM017250). The relative expression levels were calculated using the 2^−ΔΔCt^ method (Livak and Schmittgen, 2001).

### 2.6. Functional Analysis of SfUAP

To further investigate the biological functions of *SfUAP*, RNAi was performed by injecting *S. furcifera* nymphs with sequence-specific dsRNAs. First, GSPs containing the T7 promoter sequence at the 5′ end (Table 2) were used to synthesize dsRNA. Next, the plasmid DNAs of *SfUAP* and *GFP* were used to synthesize templates for in vitro transcription reactions via PCR using primers. To determine the specificity, PCR products were subcloned and sequenced. The expected fragments were then purified using the EasyPure^®^ Quick Gel Extraction Kit (Transgen Biotech). Using a Nanodrop 2000 spectrophotometer, the concentration and purity of the purified products were measured, and the integrity of the products was examined on 1% agarose gel (Thermo Fisher Scientific, Wilmington, DE, USA). Then, in vitro transcription was performed using these products.

The dsRNA was synthesized using the MEGAscript^®^ RNAi Kit (Ambion, Carlsbad, CA, USA) according to the manufacturer’s instructions. In vivo RNAi was performed in *S. furcifera* nymphs as previously described [40,41]. First-day fifth-instar nymphs were used for microinjection after being anesthetized with carbon dioxide for approximately 30 s. Each group included 50 nymphs, and treatments were performed in triplicates. Overall, 100 ng of dsRNA was injected into the nymphs between the prothorax and mesothorax using the Nanoliter 2010 Injector (World Precision Instruments, Sarasota, FL, USA). Next, equivalent volumes of GFP dsRNA (dsGFP) were used for control injections. Until eclosion, injected nymphs were reared on fresh rice seedlings as described previously. After that, the phenotype and mortality were observed daily. Photographs were captured using a Keyence VH-Z20R stereoscopic microscope (Keyence, Osaka, Japan). Subsequently, 10 nymphs were selected randomly from each replicate for qPCR.

### 2.7. Statistical Analysis

Statistical analysis was conducted using SPSS 13.0 software (IBM Inc., Chicago, IL, USA). Data values are expressed as the mean ± S.E. of three replicates. One-way analysis of variance followed by Duncan’s multiple range test (*p* < 0.05), was used to calculate the relative expression of each sample. For RNAi experiments, significant differences in mRNA levels between dsRNA treatment and dsGFP groups were analyzed using *t*-test.

## 3. Results

### 3.1. Identification and Characterization of SfUAP

The full-length cDNA of *SfUAP* was obtained from DNA fragments amplified using PCR and 5′-RACE (GenBank accession no. MF964941). The cDNA sequence of *SfUAP* is 2229 bp in size with an ORF of 1470 bp, a 5′ noncoding region of 105 bp, and a 3′ noncoding region of 654 bp, which encoded a protein of 489 amino acid residues. Figure 1 presents the complete nucleotide and deduced amino acid sequences of *SfUAP*. Using the ProtParam Server, the molecular protein formula was determined to be C_2433_H_3845_N_679_O_737_S_21_, with a theoretical molecular weight of 55.07 kDa and an isoelectric point (pI) of 6.37. Further analysis revealed that the protein lacks a signal peptide and transmembrane region, indicating that SfUAP is a cytosolic protein. Residues at the substrate-binding site that aligned with other known insect UAP proteins (Met108, Gly110–111, Met165, Gln197, Pro220, Gly222, Asn223, Ser251, Val252, Leu289, Gly290, Glu303, Tyr304, Asn327, Phe383, and Lys407) were found to be conserved; these are indicated using boxes in Figure 1. The conserved UAP motif (GGXXTXXGXXXPK; X indicates any amino acid residue) was also identified in SfUAP.

SWISS-MODEL homology modeling revealed that SfUAP comprises three domains (Figure 2): an *N*-terminal domain with seven α-helices in the *N*-terminus and one β-pleated sheet inserted into the main domain; a central domain with the Rossmann fold that consists of eight β-pleated sheets sandwiched by eight α-helices, also known as the catalytic domain; and a C-terminal domain with two β-pleated sheets and two α-helices connected to the main domain.

### 3.2. Homology Comparison and Phylogenetic Analysis

Multiple amino acid sequence alignments revealed that SfUAP shared significant sequence homology with other insect UAP proteins, such as 97% identity to *Nilaparvata lugens* UAP (AEL88647) and 69% and 63% identity to *L. migratoria* UAP1 and UAP2 (AGN56418 and AGN56419), respectively. Based on the amino acid sequences of yeast, nematode, acarina, insect, and mammalian UAPs, a phylogenetic tree was constructed using MEGA 6.06 (Figure 3). The results revealed that the *S. furcifera* UAP first clustered with hemipteran UAP, and then with UAPs from other organisms.

### 3.3. Expression of SfUAP at Different Developmental Stages and Tissues

We assessed the *SfUAP* expression profile at various developmental stages from the egg to the adult nymph (Figure 4). The results revealed that *SfUAP* was expressed in the 15 developmental stages examined. In embryonic stages, a high expression of *SfUAP* was found early (day 1). In other stages, the relative transcript level of *SfUAP* was high immediately after each centrifugation and then decreased rapidly. One-day-old adults showed the highest levels of *SfUAP* expression, whereas 2-day-old fifth-instar nymphs showed the lowest levels.

Transcript levels of *SfUAP* were evaluated in five tissue samples from fifth-instar nymphs and female adults of *S. furcifera* (Figure 5). Integument tissues showed the highest *SfUAP* transcript level, followed by the ovary, gut, and fat body tissues, whereas the head tissues showed the lowest level.

### 3.4. Functional Analysis of SfUAP

To understand the physiological function of *SfUAP*, the specific dsRNA of *SfUAP* was prepared in vitro and injected into 1-day-old fifth-instar nymphs. Total RNA was isolated from these insects 72 h after dsRNA injection, and the silencing efficiency of the target gene was detected using qPCR. As shown in Figure 6, the transcript level of *SfUAP* was downregulated, and the expression of the target gene was reduced by 89.2% (Figure 6).

The mortality rates of insects injected with *SfUAP* dsRNA were continuously monitored (Figure 7). A significant downward trend in mortality rate was observed at 12 h after injection. Furthermore, 84% of the individuals died before eclosion. Finally, 16% of nymphs underwent molting to become adults, of which 2.7% exhibited abnormalities and died.

Three malformed phenotypes were observed for the nymphs with fifth-instar individuals injected with ds*UAP* (Figure 8). First, the old cuticle was markedly split, and although the new cuticle was visible, the whole-body was unable to be released. Moreover, the abdomens of these insects were extremely shrunken or twisted. Second, the wings were not fully unfolded, and the old cuticles could not be shed off their bodies completely. Third, the new cuticle of these stunted adults was not well-hardened and looked transparent. Moreover, their wings were abnormal. All the malformed adults finally died.

## 4. Discussion

UAP is one of the crucial regulators in chitin biosynthesis during the growth and metabolism of insects. Furthermore, UAP is essential for the glycosylation of proteins and sphingolipids; glycosylphosphatidylinositol linker biosynthesis and secondary metabolites with *N*-acetylglucosamine; and the conjugation of 7-β-hydroxylated bile acids [36,42,43]. However, UAP has not been well studied, especially in hemipterous insects. Here, we report the molecular and functional characterization of the *UAP* gene from a serious rice pest, *S. furcifera**,* for the first time.

We determined the full-length cDNA sequence encoding UAP from *S. furcifera*. The *SfUAP* cDNA encoded a protein of 489 amino acid residues, with a slightly acidic pI. Similar results have also been shown in *L. migratoria* [25], *B. dorsalis* [27], and *C. medinalis* [32]. We presume this may be conducive to its function. Similar to the UAPs of other insects, SfUAP contains 17 highly conserved amino acid residues, which may be vital for substrate binding [25,27,28]. Protein sequence alignment across different organisms revealed that SfUAP and NlUAP share 97% sequence identity. This degree of amino acid sequence conservation is markedly higher than that of UAP orthologs from other organisms. Further, phylogenetic analysis revealed that UAPs from *S. furcifera* and other hemipteran insects are first clustered.

The stage-dependent expressions of the *SfUAP* transcript were investigated from the egg to adult stages. Our data show that the transcript levels of this gene were markedly different during the development of *S. furcifera*. Higher expression was detected in 1-day-old adults, suggesting that their expression correlates with an increased chitin requirement during eclosion. Furthermore, *SfUAP* expression significantly increased after each nymph molting, decreased during the inter-molting phase, and increased again before the next molt, which may be associated with nymph growth. This phenomenon was observed in *L. migratoria* and *B. mori*, in which UAP expression was periodically repeated at each molting cycle [25,34]. Additionally, we noted that the expression pattern of *SfUAP* was similar to that of chitin synthase, which is responsible for chitin formation in *T. castaneum*, *Ostrinia furnacalis*, *N. lugens*, *B. mori*, and *S. furcifera* [40,41,44,45,46]. Similar results were reported in *S. exigua*, in which the expression of *SeUAP* was high at the egg stage, L22 (2-day-old second-instar larvae), L31, L42, L51, P0 (the day after pupation), P5, P7, and A3 (3-day-old adults), correlating with a high demand for chitin biosynthesis [29]. A study on *B. dorsalis* indicated that although *Bd*G6PI and *Bd*UAP were expressed at all developmental stages, their levels were significantly higher in 1-day-old adults [27]. Similarly, in *T. castaneum*, the expression of both *TcUAP1* and *TcUAP1* was detected at all developmental stages, including trace amounts at the embryonic stages [28]. Overall, the expression profile of *UAP* in *S. furcifera* indicates that it is crucial for insect growth and development.

We further determined the tissue-dependent expression pattern of *SfUAP* using qPCR. The highest expression of *SfUAP* was detected in the integument. These findings are consistent with those in *S. exigua*, in which *SeUAP* is highly expressed in the integument and ovary, but not in the malpighian tubules and fat body [29]. Similarly, *UAP1* from *T. castaneum* and *L. migratoria* is mainly expressed in chitin-containing structures, such as the integument and gut [25,28]. However, a recent study on *B. dorsalis* showed that *BdG6PI* and *BdUAP* were highly expressed in the integument, and the highest levels of the transcript were found in the malpighian tubules [27]. Moreover, we discovered that *SfUAP* was substantially expressed in the ovaries. A similar expression pattern for chitin synthase was noted in *Mythimna separata* and *S. furcifera* [23,41]. In another study, chitin was shown to be expressed in the ovaries of *A. aegypti*. Additionally, *SfUAP* was expressed in other chitin-containing tissues, such as the fat body, gut, and head. The different expression pattern of *SfUAP* indicates that although this gene plays a major role in chitin biosynthesis, it may have additional functions in other pathways due to its expression in various tissues.

To ascertain the physiological roles of *SfUAP*, RNAi was used to silence the target gene in fifth-instar nymphs. When first-day fifth-instar nymphs were injected with dsRNA, the relative expression of *SfUAP* was significantly suppressed. The RNAi-mediated inhibition of the *SfUAP* transcript eventually affected the growth and development of nymphs and led to high mortality. Our results are consistent with those found in other insects. In *B. dorsalis*, injection of *BdUAP* dsRNA into the third-instar larvae significantly decreased the expression of *UAP*, leading to a lethal phenotype [27]. In *L. migratoria*, silencing *LmUAP1* in the nymphs caused 100% insect mortality [25]. In *S. exigua*, injection of *SeUAP1* dsRNA into the fifth-instar larvae disrupted the larval–pupal transition and led to the production of deformed pupae [29]. Thus, the physiological function of *UAP* in the nymph or larva molting is conservative in different insects. In adult beetles, RNAi for either *UAP* led to the cessation of oviposition and depletion of the fat body, leading to high mortality [28]. Thus, *UAP* may play roles that are unrelated to chitin biosynthesis in insects, such as protein glycosylation or secondary metabolite; moreover, whether *SfUAP* plays similar physiological roles in *S. furcifera* remains to be elucidated. Interestingly, in our earlier studies, similar phenotypes were observed in *S. furcifera* after RNAi silencing *SfCHS1* [41]. Altogether, these findings suggest that *SfUAP* and *SfCHS1* play interactive and essential roles in chitin formation and are important for the ecdysis and development of *S. furcifera*. This finding is supported by the findings of previous studies where silencing *SeUAP* reduced the expression levels of downstream genes (chitin synthase genes: *SeCHSA* and *SeCHSB*), particularly *SeCHSB* [29]. In summary, our results revealed that the knockdown of *SfUAP* can cause high mortality in *S. furcifera*, suggesting that *SfUAP* is a candidate gene for developing biological control measures against *S. furcifera*.

## 5. Conclusions

We cloned and characterized *SfUAP*, which is involved in the chitin biosynthesis pathway, from *S. furcifera*. During insect development and molting, this gene exhibits a distinctive expression pattern. Tissue-dependent expression indicated that *SfUAP* is mainly expressed in the integument. RNAi-based gene silencing decreased the expression levels of the target gene, led to malformations, and killed most insects. Overall, our findings not only shed light on the functions of *UAP* in the planthopper, but also provide a potential target for RNAi-based *S. furcifera* control.

## Figures and Tables

**Figure 1 genes-13-01340-f001:**
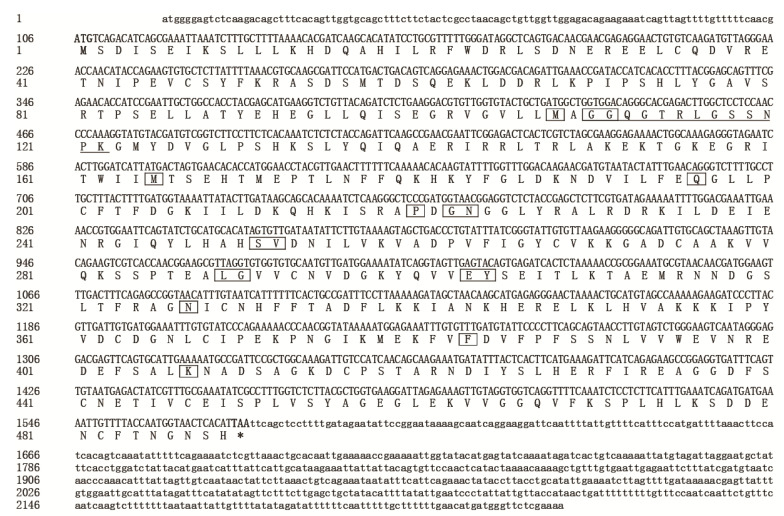
Nucleotide and deduced amino acid sequences of *SfUAP* cDNA from *S. furcifera* (MF964941). The start codon (ATG) is highlighted in bold and the stop codon (TAA) is highlighted in bold with an asterisk. The substrate-binding sites are boxed. The conserved motifs are underlined.

**Figure 2 genes-13-01340-f002:**
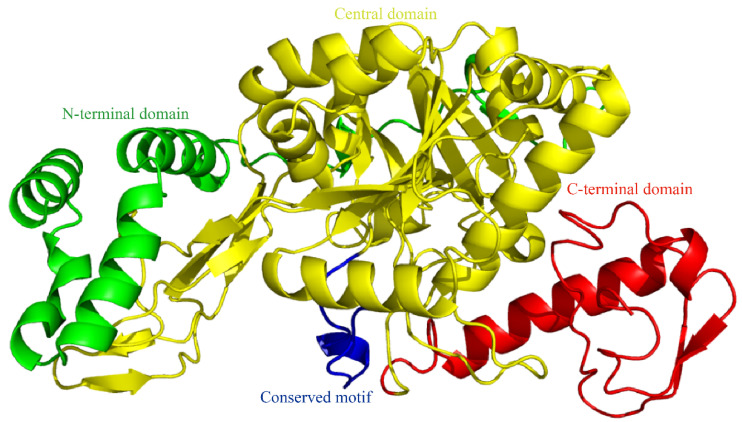
Three-dimensional (3D) model of SfUAP. The 3D model predictions generated via SWISS-MODEL homology modeling were performed using UDP-*N*-acetylglucosamine pyrophosphorylase (PDB entry code 1vm8.1.A) as templates.

**Figure 3 genes-13-01340-f003:**
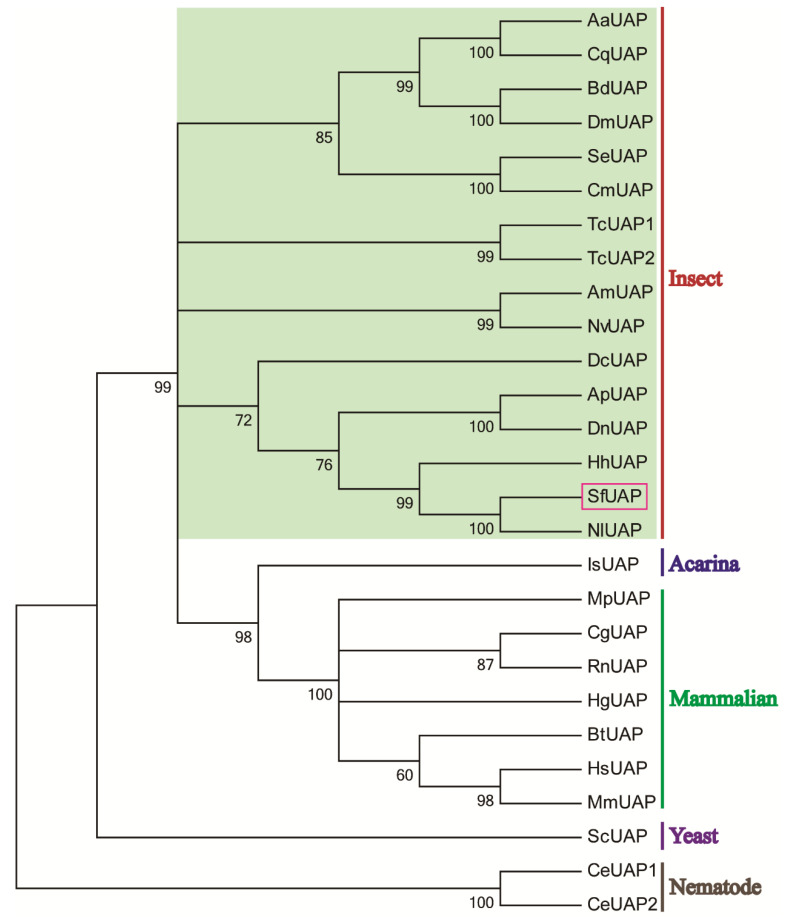
Phylogenetic tree of UAPs in insects and other organisms. The tree was constructed using MEGA 6 via the neighbor-joining method. Bootstrap analyses of 1000 replicates were performed; bootstrap values are shown next to the branches. Organisms with the associated GenBank accession numbers are as follows: AaUAP from *A. aegypti* (EAT47260), CqUAP from *Culex quinquefasciatus* (EDS38218), BdUAP from *B. dorsalis* (AGE89783), DmUAP from *D. melanogaster* (NP_609032), SeUAP from *S. exigua* (ACN29686), CmUAP from *C. medinalis* (AK090063), TcUAP1 from *T. castaneum* (NP_001164533), TcUAP2 from *T. castaneum* (NP_001164534), AmUAP from *Apis mellifera* (XP_624349), NvUAP from *Nasoniavitripennis* (XP_001602623), DcUAP from *Diaphorinacitri* (XP_008487541), ApUAP from *Acyrthosiphonpisum* (XP_001944680), DnUAP from *Diuraphisnoxia* (XP_015363117), HhUAP from *Halyomorphahalys* (XP_014289230), NlUAP from *Nilaparvatalugens* (AEL88647), IsUAP from *Ixodesscapularis* (EEC12000), MpUAP from *Mustelaputorius* (AES09131), CgUAP from *Cricetulusgriseus* (EGW06170), RnUAP from *Rattusnorvegicus* (NP_001178859), HgUAP from *Heterocephalusglaber* (EHB12810), BtUAP from *Bostaurus* (NP_001039869), HsUAP from *Homo sapiens* (NP_003106), MmUAP from *Macacamulatta* (NP_001253838), ScUAP from *Saccharomyces cerevisiae* (NP_010180), and CeUAP1 and CeUAP2 from *Caenorhabditis elegans* (NP_487777 and NP_500511, respectively).

**Figure 4 genes-13-01340-f004:**
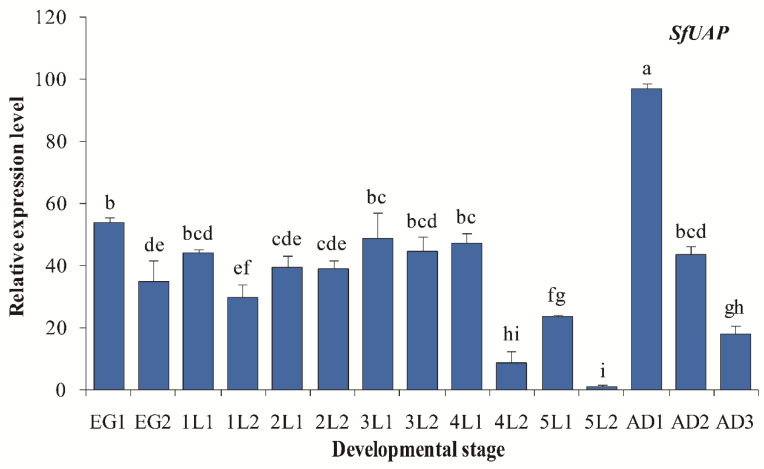
Expression profiles of *SfUAP* at different developmental stages of *S. furcifera*. At 15 different time points, expression levels in eggs, nymphs (from first- to fifth-instar nymphs), and adults were determined using qPCR. The *S. furcifera* 18S rRNA gene was used as the internal reference. Relative expression was determined based on the value of the lowest expression, which was arbitrarily set to one. Data are represented as means ± S.E. of three biological replicates. EG1, first day of the eggs; lL1, first day of first-instar nymphs; AD1, first day of adults. Different letters above the error bars indicate significant differences at *p* < 0.05.

**Figure 5 genes-13-01340-f005:**
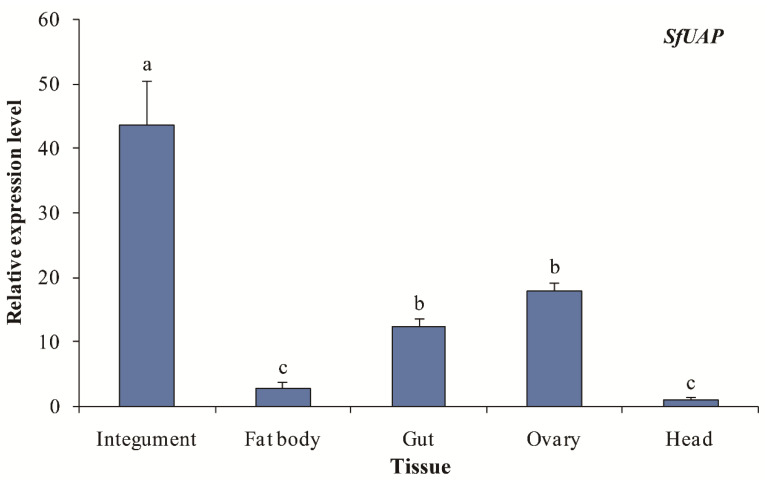
Expression profiles of *SfUAP* in various tissues of *S. furcifera*. The *S. furcifera* 18S rRNA gene was used as the internal reference. Relative expression was evaluated based on the value of the lowest expression, which was arbitrarily set to one. Data are represented as means ± S.E. of three biological replicates. Different letters above the error bars indicate significant differences at *p* < 0.05.

**Figure 6 genes-13-01340-f006:**
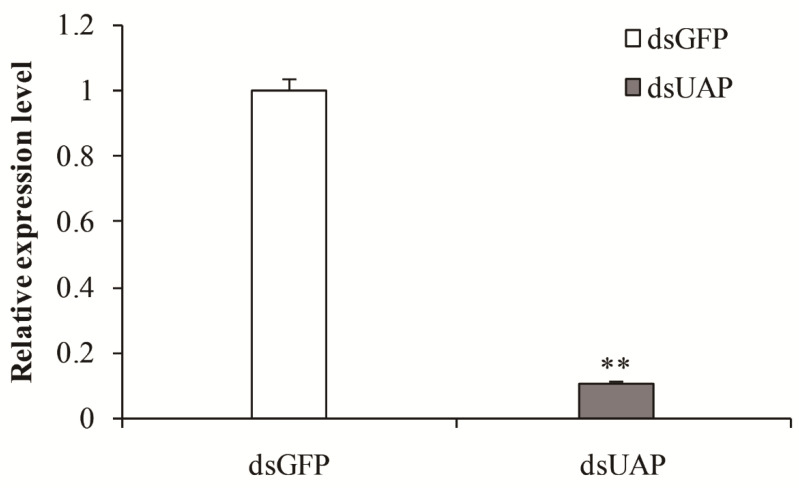
Relative transcript levels of *SfUAP* after specific RNAi treatment. The *S. furcifera* 18S rRNA gene was used as the internal reference. Data are represented as means ± S.E. of three biological replicates. ** (*p* < 0.01, *t*-test) indicates significant differences between treatment and control.

**Figure 7 genes-13-01340-f007:**
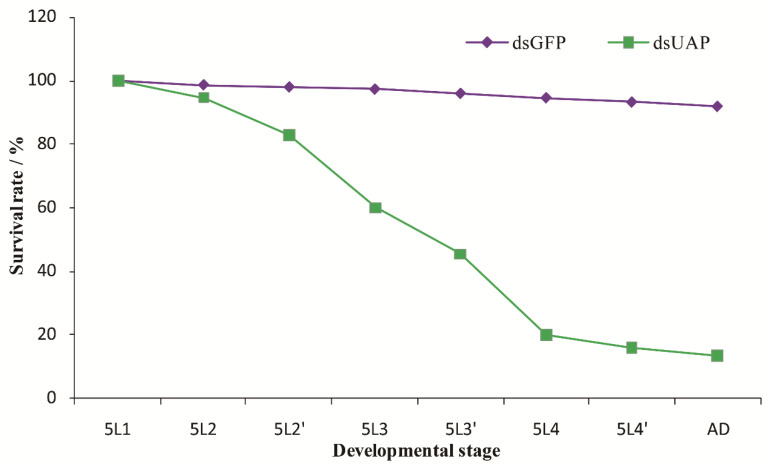
Survival rates after injection with *SfUAP* dsRNA. The survival rate of the first-day fifth-instar nymphs following the injection with *SfUAP* dsRNAs (100 ng of dsRNA was injected into each nymph). Insect age is indicated in days; e.g., 5L1 represents the first day of fifth-instar nymphs; 5L2 and 5L2′ represent the first and second halves (12 h) of a day, respectively; A.D., adults. Data are represented as mean ± S.E. from three biological replicates with fifty insects in each group.

**Figure 8 genes-13-01340-f008:**
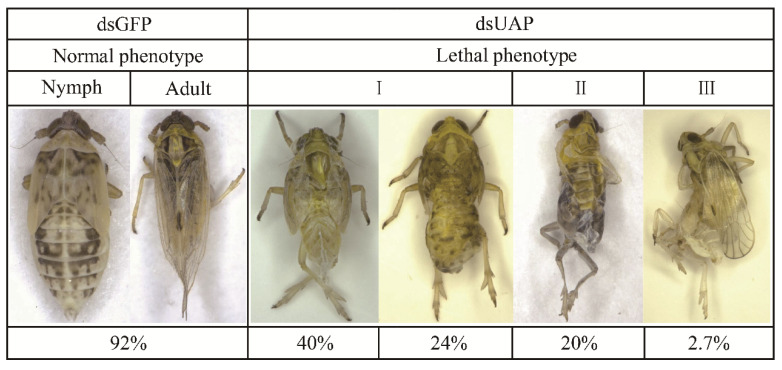
Representative phenotypes of *S. furcifera* after injection with *SfUAP* dsRNA.

**Table 1 genes-13-01340-t001:** Primers used for cloning *SfUAP* genes of *S. furcifera*.

cDNA Fragment	Primer Name	Primer Sequence (5′–3′)	Size (bp)
PCR1	UAP-F1	GAACGAGAGGAACTGTGT	770
	UAP-R1	GTTGGTGACGACTTCTGT	
5′RACE	UAP-51	CCGTAAAGGTGTGATGGTAT	335
	UAP-52	CATCTTGACACAGTTCCTCT	217
ORF confirmation	UAP-F	GTTTTTCAACGATGTCAGAC	1490
	UAP-R	GGAGCTGAATTAATGTGAGTT	

**Table 2 genes-13-01340-t002:** Primers used for qPCR analysis and dsRNA synthesis.

Experiments	Gene Name	Primer Name	Primer Sequence (5′–3′)	Size (bp)
qPCR analysis	*SfUAP*	qUAP-F	CAGCAGTAACCTTGTAGTCT	179
		qUAP-R	CGCAAACGATAGTCTCATT	
	18S *RNA*	q18S-F	CGGAAGGATTGACAGATTGAT	151
		q18S-R	CACGATTGCTGATACCACATAC	
dsRNA synthesis	*SfUAP*	dsUAP-F	TAATACGACTCACTATAGGGCGAGAACACCATCCGAAT	442
		dsUAP-R	TAATACGACTCACTATAGGGTAGAGACCTCCGTTACCAT	
	*GFP*	dsGFP-F	TAATACGACTCACTATAGGGAAGGGCGAGGAGCTGTTCACCG	707
		dsGFP-R	TAATACGACTCACTATAGGGCAGCAGGACCATGTGATCGCGC	

## Data Availability

The data were deposited in GenBank with accession number MF964941.

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
