# Peer review of "Molecular Characterization of UDP-N-Acetylglucosamine Pyrophosphorylase and Its Role in the Growth and Development of the White-Backed Planthopper Sogatella furcifera (Hemiptera: Delphacidae)"

_genes, 2022, doi:10.3390/genes13081340_

Round 1
Reviewer 1 Report
Dear Authors
Thank you for your submission. Article still needs revision before consideration.
Authors are currently unable to provide the complete and chronological text for the designed study. Please also make the objective more clear.
How authors are sure that these collected insects are pesticides free? Very important to discuss...... because chemicals directly effect the physiology of insects and ultimately molecular level of insects
Rearing should be described in detail.... Along with their generations for the experimentation.
As authors have used the post hoc test, where the treatment groups and their description?
Results are in better position.
Discussion and conclusion must be improved. Very less number of references have been used in the discussion, which is not proven a strong discussion.
Other comments are present in the attached file.

Reviewer 2 Report
Authors isolated cDNA , 5’ and 3’ UTR segments of UDP-N-acetylglucosamine pyrophosphorylase from white-backed planthopper. Research is interesting in context of metabolic functions of this enzyme and potential role in regulating the white-backed planthopper population. Research is properly planned and performed, results support conclusions. Some minor comments should be addressed before the publication.
1. Page 2, sentence in line nr 3 (starting from the top of page) begins with a small u letter- capitalize it.
2. Page 2, sentence in line nr 4 (starting from the top of page)- should be trehalose.
3. Section 2.3
Authors should provide more details concerning the 5’RACE reaction.
Putatively the 770 bp fragment obtained after PCR1 reaction contains the 3’UTR segment. Authors should state it more clearly in section 2.3.
4. Section 2.4
Authors wrote that used two different development stages to isolate RNA and perform RT-PCR and finally characterize the organ-specific gene expression. However, the Fig. 5 shows not results for these two developmental stages, Authors should correct it.
If Authors mixed RNA from these two stages it should be explained why they did so and why selected just these two stages.
Amount of RNA taken for 1 sample.
Was the isolated RNA digested by DNase to remove remnants of genomic DNA?
Round 2
Reviewer 1 Report
Dear Authors
No substantial change was found in the introduction and Discussion except citation improvements. Authors must take it seriously.
Citation 1-6 must be separately cited where applicable.
New modified text should be red in color.
Author Response
Please see the attachment。
